# CEDe: A collection of expert-curated datasets with atom-level entity annotations for Optical Chemical Structure Recognition

**Rodrigo Hormazabal[1,2], Changyoung Park[1], Soonyoung Lee[1],**
**Sehui Han[1] Yeonsik Jo[1], Jaewan Lee[1], Ahra Jo[1]**
**Seunghwan Kim[1], Jaegul Choo[2], Moontae Lee[1], Honglak Lee[1]**
[1]LG AI Research, [2]KAIST
{rodrigo, changyoung.park, soonyoung.lee, hansse.han
yeonsik.jo, jaewan.lee, ahra.jo
sh.kim, moontae.lee, honglak}@lgresearch.ai
jchoo@kaist.ac.kr

## Abstract

Optical Chemical Structure Recognition (OCSR) deals with the translation from chemical images to molecular structures, this being the main way chemical compounds are depicted in scientific documents. Traditionally, rule-based methods have followed a framework based on the detection of chemical entities, such as atoms and bonds, followed by a compound structure reconstruction step. Recently, neural architectures analog to image captioning have been explored to solve this task, yet they still show to be data inefficient, using millions of examples just to show performances comparable with traditional methods. Looking to motivate and benchmark new approaches based on atomic-level entities detection and graph reconstruction, we present CEDe, a unique collection of chemical entity bounding boxes manually curated by experts for scientific literature datasets. These annotations combine to more than 700,000 chemical entity bounding boxes with the necessary information for structure reconstruction. Also, a large synthetic dataset containing one million molecular images and annotations is released in order to explore transfer-learning techniques that could help these architectures perform better under low-data regimes. Benchmarks show that detection-reconstruction based models can achieve performances on par with or better than image captioning-like models, even with 100x fewer training examples.

## 1 Introduction

The recognition of molecular structures presented as images in scientific literature is an essential part of material design and drug discovery pipelines [1, 2]. Most chemical structures are typically shown only as 2D structural depictions in image form instead of digital representations using parsable data formats. The lack of a widely adopted standard to publish molecular structures slows down the material discovery process and poses a long-standing problem in effectively searching for seemingly unexplored parts of chemical space in previous literature. This problem worsens when it comes to patents containing newly discovered chemicals or drugs. Chemical structure descriptions tend to be superficial, ignoring certain structural details and common conventions, while in most cases still being solely presented in image format [3]. Naturally, this leads to the introduction of uncommon symbols and markers on chemical structure images, making them even harder to be discovered. Therefore, transforming and storing existing literature and molecular structures as indexable data formats is a critical challenge to improving current chemical databases.

36th Conference on Neural Information Processing Systems (NeurIPS 2022) Track on Datasets and Benchmarks.

Chemical information in scientific literature is commonly presented in various ways, such as text, tables, charts, and images. In the case of images, 2D projections of chemical structures are the de facto standard to represent compounds in literature. Currently, the majority of the chemical structure images in scientific documents are not machine-readable, and the automatic identification of these structures is still a challenging task. This task, traditionally known as **O**ptical **C**hemical **S**tructure **R**ecognition (OCSR), entails the recognition of chemical structures from their image form depictions and the generation of their corresponding machine-readable representations, such as InChI [4] or SMILES [5].

In the past, various methods have been proposed for the task of chemical structure recognition. Rule-based methods based on segment detection and image vectorization have been developed by several research groups, such as Imago [6], Optical Structure Recognition Application (OSRA) [7], and MolVec [8]. Most of these work has followed similar pipelines; (1) detection of lines and their intersections using a series of classical computer vision heuristics and techniques, followed by (2) the reconstruction of the molecular graph via linking the recognized structural components. These methods mostly work well for shallow structures that do not deviate far from commonly seen molecular patterns. However, they tend to fail at recognizing more complex images perfectly, for example, ones containing stereoisomers. Correctly identifying complex structures entails not only detecting molecular constituents but also capturing implicit information within the underlying molecular graph connectivity. These approaches, primarily based on hand-crafted rules, tend to quickly hit a hard wall in terms of performance, constantly struggling with less common yet critical structural details.

Recently, many efforts have been put into tackling OCSR from a data-driven perspective, trying to leverage modern computer vision techniques based on deep learning [3, 9, 10]. Most of these methods follow pipelines analog to image captioning, directly translating raster images containing structures into their molecular string representations such as InChI, SMILES, or SELFIES [11]. Due to the high availability of chemical databases containing millions of these string representations, such as ChemBL [12] and PubChem [13], previous approaches generate large training datasets of synthetic images[3, 10] using chemoinformatics toolkits, like `RDKit` [14] and `Indigo` [15].

However, even with these large synthetic datasets, existing approaches show performances at most on par with the traditional rule-based methods. There are two main reasons why these models fall short of improving over traditional techniques and lack generalization power. First, the recurrent nature of the string generation process leads to loss functions that are not entirely aligned with the task of structure recognition as a whole, ignoring important inductive biases present on molecular graphs. Chemical entity interactions that occur naturally on graph-like structures must be learned directly from pixel information, rendering these models highly data-inefficient [10]. Second, long interaction patterns between atoms are common in aromatic compounds and stereoisomers, where the coordinated interaction between possibly long chains of atoms can induce special behavior within compounds themselves. Aromaticity involves conjugated bonds with delocalized electrons, which tend to span across rings of atoms, and to efficiently identify these phenomena, each atom's graph neighborhood information is necessary. Stereoisomers have the same constituent chemical entities; however, they differ in how atoms are spatially arranged. This can have substantial implications regarding molecular behavior, and reasoning across distant parts of the graph structure might be necessary to identify them correctly. These long-range interactions are challenging to learn in string space. On the contrary, these interactions are far easier to propagate and infer on graphs while also being more stable to train. A data-driven chemical entity recognition step followed by a graph construction approach, similar to traditional methods, shows itself as a better-suited alternative to incorporate the chemical structure's inherent graph properties.

The main advantage of approaching the OCSR task through a detection pipeline is that multiple instance training examples are available per image, creating a fine-grained learning signal for the backbone model and overall improving data-efficiency compared with direct image-to-SMILES methods. However, no open-source dataset containing this sort of annotations is currently available, either synthetically generated or extracted from real documents. In our opinion, this has been the main limiting factor for the lack of exploration of this kind of approaches.

We present a collection of chemical entity bounding boxes manually curated by experts for scientific literature datasets, named "**C**hemical **E**ntity **De**tection" (`CEDe`), that looks to encourage research on approaches that follow the aforementioned strategy: pipelines that combine data-driven computer

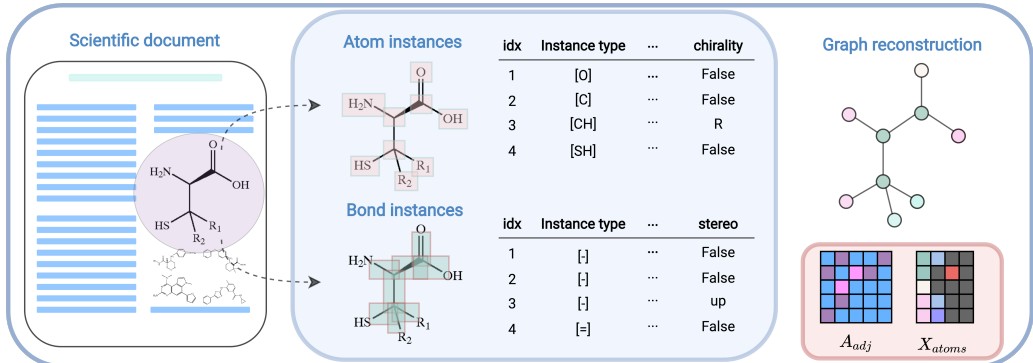

Figure 1: Main pipeline for Optical Chemical Structure Recognition based on atom-level molecular entity detection and graph reconstruction. Atoms, bonds, and other instances that compose a molecular graph are detected and subsequently combined to construct the underlying compound structure. Information related to the stereochemistry of each instance is necessary to disambiguate isomers. All this information is provided for all the instances in the CEDe datasets.

vision architectures with molecular compounds' inherent graph structure. To allow for these kinds of architectures to be developed, atom-bond level positional information within the image and their corresponding chemical information must be available. Following the detection of these molecular entities, complete molecular structure reconstruction is possible in a way that is better aligned with the task itself. We show that unlike image-captioning based OCSR pipelines, formulating this task as an entity detection-graph reconstruction problem can produce efficient models that can learn with far fewer examples (in the order of tens of thousands instead of millions).

CEDe consists of a set of more than 700,000 bounding box annotations with carefully designed labels for chemical entity identification and molecular graph reconstruction across four real scientific documents OCSR datasets. These bounding box annotations were manually curated by experts while considering conflicting edge cases to design an adequate set of labels for each chemical entity type. In addition, a synthetic dataset is also released to motivate the exploration of pre-training strategies for chemical entity detection architectures. The synthetic version of CEDe includes the necessary rendering style-related augmentations to deal with the vast diversity present in scientific documents. Also, we release the codebase used to generate this dataset, hoping to allow researchers to either pre-generate new datasets designed for a particular use case or explore adaptive style-augmentation strategies as a part of the training process.

The contribution of our work and the presented datasets can be summarized as,

- We design a chemical-entity label set (atoms, bonds, charges, pseudoatoms, etc.) that covers all the necessary information for compound structure identification from its constituents. These labels allow for the reconstruction and disambiguation of complex cases, such as stereoisomers, which need information that might not be explicitly present on each molecular image.

- Following the presented label set, we annotate and release four of the most widely used OCSR datasets with bounding box annotations for chemical-entities and its corresponding labels (UOB [16], USPTO [17, 18], CLEF [19], JPO [20, 21]). These datasets were manually curated by domain experts and contain more than 10,000 images and 700,000 instance-level annotations.

- We release a synthetic dataset containing one million molecular images with atoms and bonds positional information within the image and their respective labels. In addition, a carefully designed set of rendering style augmentations is provided in order to match the style diversity present in chemical documents and help generalization from synthetic to real data (fonts, font sizes, line thickness, atom-orientation preserving rotations, etc.)

- We release a codebase for synthetic data generation that includes all the aforementioned style augmentations that can also be efficiently parallelized to generate data while training and exploring adaptive augmentation setups.

- We run baseline benchmarks for the currently most used architectures for OCSR, which are based on image-to-SMILES translation. In addition, we present and benchmark pipelines that can leverage the new annotations presented here, named detection-reconstruction approaches. We run experiments testing synthetic-to-real performances by pretraining on our generated datasets and also fine-tuned performances by using a small subset of the CEDe annotations for training.

## 2  Existing OCSR datasets

Most recent work on deep learning-based approaches for OCSR [3, 10, 22, 23, 24] use pipelines based on architectures commonly used for image captioning, i.e., extract features from a chemical structure image, which are then fed into an autoregressive decoder module that generates a string-representation for each compound. In order to train this kind of models, previous approaches either use synthetic datasets generated with chemoinformatics libraries, or use datasets containing labeled images extracted directly from scientific documents and patents.

Labeling images extracted directly from scientific literature is a time-consuming process that has to be done by domain-expert annotators due to the complex nature of the task. As a result, there is a restricted amount of real datasets containing chemical structure annotations, such as MOL files (Chemical Table Files) or SDF files (Spatial Data File). The most well-known and widely used datasets are listed in Table 1; (1) UOB is a dataset developed by University of Birmingham, United Kingdom, (2) USPTO dataset is composed of structures appearing on patents from the Unites States Patent and Trademark Office, (3) CLEF was developed from the Conference and Labs of the Evaluation Forum, and (4) JPO was collected from the Japanese patent office documents. A detailed analysis on these datasets and their characteristics can be find in a previous work by Rajan et al. [25].

These openly available datasets combine to around 12,000 annotated images, which is not enough to successfully train end-to-end models using modern computer vision pipelines based on deep learning. Due to this, previous work tackling OCSR has resorted to synthetic data generation approaches. In contrast to the expensive image annotation process, synthetic data generation is fast and flexible. Image annotations can be generated at any level of complexity, including atom/bond exact positions within the image, labels regarding stereochemistry information, and string representations of the compounds. This kind of annotations can be more informative to data-driven models and, allow for the use of less explored architectures in OCSR, such as the integration of detection models into the structure recognition pipeline. However, designing the necessary augmentations to allow for synthetic-to-real generalization shows to be a complex task, which is one of the main limitations when relying solely on synthetics samples. On the other hand, labeling of atom/bond level annotations for real images can be even more expensive than overall molecular level labels and, currently there is no widely used standard for this annotation scheme. As a result, real datasets containing atoms-bonds positions and labels are non-existent.

Table 1: Datasets containing MOL or SDF file labels for molecular images extracted from patents and scientific documents.

| Datasets | Data samples | Source |
|---|---|---|
| UOB | 5740 | University of Birmingham [16] |
| USPTO | 5719 | Unites States Patent and Trademark Office [17, 18] |
| CLEF | 961 | Conference and Labs of the Evaluation Forums [19] |
| JPO | 450 | Japanese Patent Office [20, 21] |

# 3 Datasets

## 3.1 The CEDe dataset

`CEDe` consists of a collection of datasets containing atom-bond level instance annotations of compounds appearing on real scientific documents and patents from the four aforementioned open-source datasets; `UOB`, `USPTO`, `CLEF` and `JPO`. Bounding box annotations were manually curated by domain experts, with a carefully designed set of labels for reconstructing molecular graphs correctly when compared to the available ground truth annotations. The structure ground truth information can be found as `MOL/SDF` files for all compounds in the open-source datasets and is used to verify missed or mislabeled instances in each chemical image.

Figure 2: Example of complex structures present in scientific literature. **(a-b)** `CLEF`, **(c-d)** `USPTO`, **(e-f)** `UOB`, **(g-h)** `JPO`. These sort of cases strongly differ from the render styles captured by synthetic data generators and underlines one of the main challenges when training models fully on synthetic data.

**Labeling process and format**   In order to develop OCSR pipelines based on molecular entity detection, atom-bond level instances and graph connectivity information must be available, as is shown in the `CEDe` annotation example in Fig. 3. In order to provide a sanity check for human annotators, the available `MOL/SDF` files were converted into molecular graphs using `RDKit` to be utilized as reference (atoms as nodes and bonds as edges). Each chemical entity was transformed into their respective SMARTS tokens [26], which are pattern-matching strings containing atom information, such as charge and aromaticity, and chirality, as well as bond information, such as stereochemistry. This allowed annotators to know beforehand the type and quantity of each instance label that must be present within a certain molecular image.

Some structural information necessary for unambiguous graph reconstruction is not directly represented in each chemical instances' pixels and must be inferred directly from the molecular graph connectivity. This is the case of labels such as the implicit number of hydrogens, aromaticity, and chirality, which are commonly not shown in images with any explicit symbol. As for edges, stereo information can only be determined by looking at the position of the connected atoms within the image. In these cases, expert annotators could rely on information extracted from the molecular files to double-check their annotations.

**Bounding box annotations format**   `CEDe` follows the `COCO` annotation format for object detection [27], with bounding box coordinates `[x1,y1,x2,y2]` and a multi-label assigned to each instance within the image. Annotations are provided as a `JSON` file with metadata for each image, corresponding instance annotations, and a category dictionary that maps category indexes and the instance label name. Bounding boxes were drawn, keeping each instance near the center of the box and, at the same time, also considering their immediate neighbor information, such as corresponding bond directions.

**Pseudoatoms**   All pseudoatom annotations are labeled manually by annotators and classified into one of three subcategories: `"[element]"` (letters that directly map to a single atom), `"[atom_bundle]"` (chemical formulas and functional groups, e.g., CH2CH3, t-Bu, etc.), and `"[OCR]"` (arbitrary characters that do not represent explicit chemical information, e.g., R1, R2, X, Y, Q, etc.). ''[atom_bundle]'' subcategory instances can be mapped directly into a subgraph/SMARTS query (e.g., `[t-Bu] -> "CC(C)(C)[*]"`). Instances that need OCR represent cases where the text

is used as a pointer to a certain part of the graph. An example of atom and bonds bounding box annotations is shown in Fig. 3.

Detection-reconstruction baselines presented in our work handle unseen pseudoatoms by defining an instance class called "others", which represent classes not seen while training (or classes with an insufficient number of examples to learn meaningful mappings), which are then predicted with the use of OCR tools. This is just one of many possible approaches to tackle this problem, and more efficient ways could be designed. It is important to note that image-to-SMILES methods also need to deal with this problem (unseen tokens while training cannot be predicted at inference time), and solutions might be much more cumbersome since positional information about the unseen class tokens is not available.

**Generation of SMILES from detected entities**   Detected instances and bonds are used to generate the corresponding molecular graphs directly in `RDKit`, i.e., adding atoms and bonds to an RWMol object. This information is sufficient to transform any molecular graph directly into any other format, such as SMILES, `SDF`, or `MOL` files. In the case detection-reconstruction models, after detecting the chemical entities contained within an image, a module that handles connectivity prediction (bonds) needs to be implemented, which can be based on heuristics (e.g., find overlapping bond-atom bounding boxes) or learned modules (e.g., use pairs of detected atoms' bounding boxes, together with an activation map of the area between instances). With this information, SMILES strings can be generated just like explained beforehand.

**Reducing errors and noise in annotations**   Since human annotations can contain noise or errors; we perform a series of post-processing steps and tests to verify the correctness of each sample:

- Check that the set of annotated instances within an image is sufficient to reconstruct the chemical structure.

- Match the annotated bounding boxes with their corresponding SMARTS substructures contained in the ground truth chemical file (`R/S` for chirality of atoms, `E/Z` for the stereotype of bonds, labeled as `[STEREOE]`/`[STEREOZ]` following their name in `RDkit`, aromatic atoms, etc.). `MOL/SDF` files are available for all datasets labeled in this work, and we use these to double-check our annotations.

- Verify that bounding box annotation sizes and positions do not deviate far from the distribution of **(a)** other instances within an image and **(b)** same class instances in other images.

- Verify that connected bounding boxes do not violate chemistry rules, such as valency. For example, we check for possible mislabeled aromatic atoms by checking their neighbors. Even though aromatic information can be inferred directly from atoms and bonds, we purposely labeled aromatic atoms separately (following SMARTS convention, with lowercase letters) to reduce possible errors.

- All annotations that fall under the subcategories of ``[element]'', ``[atom_bundle]'', and ``[OCR]'' are also predicted with OCR tools and extracted from the available molecular files. These three sources are cross-checked against each other to detect possible mislabeled instances.

In addition to this, annotation boxes containing letters are tightened by cropping white space around them with classical computer vision techniques. A script to perform this process or leave the original human annotations is provided in our repository.

## 3.2   CEDe Synthetic dataset

We designed a synthetic dataset containing a wide range of chemical compounds to cover most of the structural diversity found in scientific literature. We paired this with a carefully designed set of augmentations for the depicted graph structure in order to deal with the domain shift between synthetic and real images. In addition, our synthetic dataset consists of molecules containing a diverse set of pseudoatom entities, structures with complex stereochemistry, and charged atoms. Most importantly, molecular constituent annotations with their corresponding labels and bounding boxes (e.g., atoms, bonds, charges) are made available to motivate the exploration of pre-training strategies for the presented pipeline based on detection modules followed by a graph reconstruction step.

Figure 3: `CEDe` image annotation example. **(a)** Chemical image example. **(b)** Structure meta-labels. Includes count of unique instances within the image and string representations like SMILES. **(c)** Instance annotations. Contains the corresponding bounding box information, category ID, SMARTS token and, specific information regarding stereochemistry.

Molecular images with their corresponding instance-level information were generated using `RDKit`. The `RDKit` chemoinformatics library is implemented in C++, and wrappers around Python are available. Instance bounding box annotations and their corresponding labels can be directly calculated as part of the image generation process while freely applying augmentations to the image styling. We sampled a subset of one million structures from the PubChem database [28], a collection of more than 100 million chemical compounds, covering a diverse group of complex structures containing all molecular entities commonly found in scientific literature. More details regarding the sampling method and label distribution can be found in the supplemental material.

**Image and structure augmentations** We do not impose any restrictions on structure size or element types in our sampling process, which is commonly done by previous approaches utilizing synthetic data generation. Chemical images were generated with a resolution of 768 by 768 pixels. In order to make the learned models more robust to different rendering styles, we developed a set of style augmentations during the data generation process. These augmentations have clear differences from augmentations commonly used in modern computer vision pipelines, such as rotation, Gaussian noise, or salt-and-pepper noise. For example, directly rotating a molecular structure image would also rotate the atom characters within the compound, which does not reflect real cases found in scientific documents. In total, six types of augmentations were randomly applied to each generated compound image, as shown in the example in Fig. 4; **(a)** bond line-thickness **(b)** random substitution of an atom entity with a random pseudo-atom (substructure abbreviated form) **(c)** shear-like XY scaling of the structure's skeleton while conserving letters style **(d)** atoms' font style **(e)** atom font size **(f)** structure rotation while preserving characters orientation.

## 4 Benchmarks

Three chemical entity detection baselines, consisting of a detection module [29, 30] and a reconstruction step [31], were benchmarked against three CNN encoder-autoregressive decoder architectures [32, 33, 34, 35]. An overall diagram of these architectures is shown in Fig.5. In order to compare the sample efficiency of the detection-based methods to direct image-to-SMILES translation approaches, we run a set of experiments with different amounts of training data for each model. In our evaluation setup, image-to-SMILES models are trained with the complete set of one million synthetic images. Training these models with fewer samples leads to weak baselines that can barely start understanding SMILES grammar rules, generating mostly invalid or incomplete sequences. For the detection-reconstruction baselines, we used a small dataset containing only 1% of the training data used on

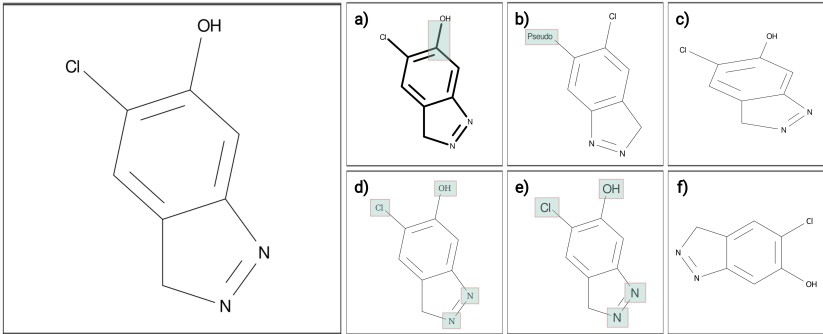

Figure 4: Default style of our rendered chemical structures and their augmented versions. (a) line-thickness, (b) pseudoatoms, (c) shear-warping, (d) font style, (e) font size, (f) structure rotation. Shear-warp and structure rotation maintain the style of atom characters.

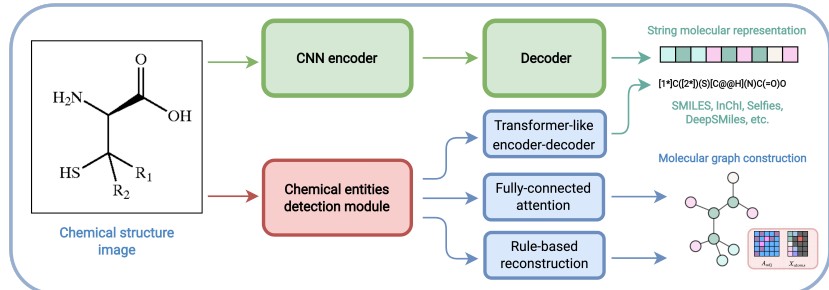

Figure 5: Main approaches to Optical Chemical Structure Recognition. CNN-decoder pipelines extract image features with a backbone network and, autoregressively decode tokens to generate a string representation of the molecule. Chemical entity detection approaches, first detect chemical-entities and subsequently derive the connectivity from the detected instances to generate the underlying chemical graph.

image-to-SMILES models. This smaller training dataset was sampled with the restriction of having a minimum number of each chemical entity type present in the original one million synthetic images. Both sets of architectures are evaluated on the same four real datasets and one extra synthetic dataset of 10,000 images generated solely for testing purposes. More details on the training set sampling strategy, validation set generation, and model hyperparameters can be found in the supplemental material. For reference, available OCSR software was also tested on the four real datasets and compared against the presented baselines.

**Fine-tuning** We also consider the case where models are pre-trained with synthetic data and fine-tuned on a subset of CEDe, i.e., real images coming from scientific documents. To accomplish this, a small subset (20%) of each of the four real datasets is combined into a different split used to fine-tune all models.

**Evaluation metrics** Baseline models' performance was evaluated through graph matching accuracy between the ground truth structures and the recognized ones. This can be accomplished by transforming graph structures into a canonical string representation, like InChI or canonical SMILES, and reducing this process to a string-matching evaluation. Also, similarity measures have been used to evaluate OCSR models, such as Tanimoto similarity (substructure similarity) [36], Levenshtein distance (string similarity) [37] and graph edit distances (graph similarity) [38]. However, we do not use any of these measures to evaluate models due to the natural imbalance within the chemical entity tokens distribution (check supplemental material). This leads most similarity metrics to ignore under-represented yet critical chemical entities, such as pseudoatoms. In order to compare the presented detection architectures, we also report each model's mean average precision (mAP) for all entity classes in the supplemental material.

Table 2: CNN-decoder and chemical entity detection baseline models benchmark results on one synthetic and four available datasets coming from real scientific documents.

| Test data | Training dataset | Rule-based methods | | | Image-to-SMILES translation @ 1M data | | | Chemical entity detection @ 10K data | | |
|---|---|---|---|---|---|---|---|---|---|---|
| | | Imago [6] | MolVec [8] | OSRA [7] | CNN+ GRU [32, 33] | CNN+ GRU+ Attention [34] | CNN+ Transformer decoder [35] | DETR+ bond bbox connectivity [29] | FasterRCNN+ bond bbox connectivity [30] | FasterRCNN+ Instance-pair Transformer [30, 31] |
| **Synthetic dataset (10K)** | Synthetic | 79.21 | 85.92 | 87.34 | 59.2 | 61.2 | 57.4 | 79.81 | **88.96** | 86.15 |
| **UOB** (4592) | No pretraining | - | - | - | 5.12 | 7.99 | 3.12 | 47.24 | 52.12 | 49.77 |
| | Synthetic | - | - | - | 32.01 | 36.37 | 34.84 | 65.59 | 74.09 | 67.94 |
| | **Fine-tuned** | 63.83 | 89.11 | 85.95 | 62.57 | 66.9 | 67.29 | 84.32 | **91.49** | 89.26 |
| **USPTO** (4575) | No pretraining | - | - | - | 4.12 | 5.32 | 2.19 | 52.91 | 58.24 | 51.99 |
| | Synthetic | - | - | - | 39.78 | 43.98 | 42.60 | 63.83 | 78.95 | 75.43 |
| | **Fine-tuned** | 86.91 | 88.13 | 87.83 | 48.52 | 55.17 | 53.51 | 86.12 | **90.97** | 88.11 |
| **CLEF** (768) | No pretraining | - | - | - | 1.43 | 1.56 | - | 48.12 | 55.29 | 47.25 |
| | Synthetic | - | - | - | 20.31 | 23.18 | 21.61 | 53.65 | 67.97 | 64.97 |
| | **Fine-tuned** | 66.41 | 82.94 | **92.84** | 35.16 | 39.19 | 36.59 | 73.31 | 86.59 | 81.64 |
| **JPO** (360) | No pretraining | - | - | - | 0.83 | 0.27 | - | 42.93 | 47.29 | 36.88 |
| | Synthetic | - | - | - | 15.28 | 17.22 | 14.17 | 30.56 | 49.44 | 45.83 |
| | **Fine-tuned** | 41.39 | 67.22 | 58.89 | 33.61 | 30.28 | 28.06 | 55.83 | **74.12** | 59.72 |

## 4.1 Synthetic data - pretraining

For the synthetic data benchmarks, image-to-SMILES models show the lowest performance across all models by a considerable margin, even with one million training examples coming from the exact same data distribution. These models need many samples just to start generating valid SMILES strings since they must learn all the nuanced rules related to the string-representation system itself. These models have an even harder time learning SMILES strings on low-data regimes, even for small and simple sequences, which can be seen in the results for models without any synthetic data pre-training. Previous work has used datasets in the order of 30 to 50 million data points to see good performances on synthetic data (>95%) [23]. As is shown in table 2, chemical entity detection-based models perform the best across all datasets with just 10,000 examples. It is important to note that the small 10K synthetic training dataset contains high chemical diversity, covering all the possible chemical entities that appear in the full synthetic test split.

## 4.2 Real data - fine-tuning

Faster-RCNN with a rule-based connectivity reconstruction model performs on-par or outperforms all other architectures (improves of 2~7%↑, except on CLEF), while only using 10,000 synthetic images and around 2,575 real images. Results on the fine-tuned performances show how pre-training with synthetic data, followed by a fine-tuning step with just a few examples coming from the test-style distribution, can greatly improve models. This is an expected result of the fine-tuning process; however, the improvements shown by models such as CNN+GRU on the JPO dataset (x2.2 times) are worth noticing. One possible explanation for these improvements is that after pre-training these models with 1M data points, the new bottleneck becomes the distribution shift between chemical-image styles and not the SMILES grammar, as it is at the beginning. Rule-based methods tend to perform similarly across most datasets (except, JPO), beating all baselines trained with only synthetic data. This hints at the very nature of expert models based on heuristics, which perform well within the scope they were designed for but tend to fail when it comes to less common structural patterns, rendering them hard to improve.

## 5 Limitations and further work

The first version of our dataset does not include Markush structures, polymeric compounds, and chemical reaction paths. The main reason behind not considering these chemical structures/systems

in our datasets' initial version is that they need a completely different treatment when it comes to reconstruction. For example, instead of using SMILES and InChI to represent chemical reactions, we might have to rely on some other system, like SMIRKS [26]. Even though we can create annotations for the constituent chemical entities, the reconstruction step must also consider the different reactions, Lewis structure notations, pathways, and the relation between regents/products. We plan to add more complex cases in future versions of our datasets.

# 6 Conclusion

In this work, we present `CEDe`, a collection of chemical instance-level annotations for OCSR tasks, consisting of bounding boxes and labels manually curated by experts for open-source scientific literature datasets, which combines to more than 700,000 chemical entity annotations. These annotations, combined with a carefully designed set of labels for chemical entities, comprise all the necessary information for unambiguous chemical structure reconstruction. In addition, a large synthetic dataset containing one million molecular images with atoms and bonds positional information with their respective labels is released. In order to combat the distribution shift from synthetic to real images, a set of carefully designed rendering style augmentations is provided. These style augmentations, together with more commonly used modern computer vision augmentation pipelines, can help to cope with the style diversity present in chemical documents and improve generalization. A codebase for synthetic data generation is also released to encourage research into transfer-learning techniques that can allow these architectures to perform better under low-data regimes, where the diversity in chemical structure rendering styles poses a challenging problem. We benchmark baseline architectures based on image-to-SMILES translation, which follows an encoder-decoder framework against detection-reconstruction and rule-based methods. Our benchmarks show that detection-reconstruction models pre-trained with synthetic data and fine-tuned with small subsets of the real `CEDe` annotations perform on par or better than all other reported baselines. Our baseline benchmarks show that this is the case even when the detection-reconstruction models use just 1% of the pre-training data used by image-to-SMILES models while showing significantly better performance.

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
