# OpenReview forum: "CEDe: A collection of expert-curated datasets with atom-level entity annotations for Optical Chemical Structure Recognition"
_NeurIPS.cc/2022/Track/Datasets_and_Benchmarks — NeurIPS 2022 Datasets and Benchmarks _

### Official Review · Reviewer_gkZt · 2022-07-14
**Very good dataset and provide a good augmentation method, but more cases need to be considered**

**Rating:** 5
**Confidence:** 4
**Correctness:** It is constructed in a sound way as a…

**Strengths:**

1. this work considers some important features in chemical structures recognition, such as stereochemistry (from both atom and bond perspective), different image styles because of different journal requirements.
2. This work did a very thorough data augmented types: bond line-thickness, substructure abbreviated form, shear-like XY scaling, atom font style and size, structure rotation (Figure 4).
3. the evaluation metrics using exact graph matching accuracy by transforming the structure into the InChI form is reasonable.
4. This work treated a very common phenomenon in chemistry, that is in literature, the chemical structures are drew with chemistry abbreviations, such as Me (means CH3), Et(CH2CH3), Boc (a protecting group). The author analyzed the distribution of pseudo-atom in four dataset. And used them for synthetic data augmentation.

**Weaknesses:**

1. the paper did not consider atoms' or functional groups' representation, such as NHBoc and BocNH, OAc and AcO, t-Bu and tBu and so on.
2. As the paper claimed it is a collection of expert-curated dataset, it will be much better to combined automating abstracting the pseudo-atoms from known dataset with expert rules to include more hard cases.
3. Besides the stereochemistry of atom, the E/Z of double bond is also needed to be considered. I don't see the discussion about that clearly.


**Additional Feedback:**

I think it's better to add more examples showing that current algorithms failed and succeeded by training with CEDe dataset.

**Clarity:**

The paper did not describe clearly how they predict the structure containing pseudoatoms, such as in CLEF dataset in Figure 2, a and b. Are they predicted as X, Y, W, Z, Ar in SMILES strings if they are not common atoms?

**Documentation:**

The documentation seems sufficient.

**Ethics:**

I have not found any ethical concerns.

**Relation To Prior Work:**

The comparison with previous methods, such as rule-based methods, are clearly discussed.

**Summary And Contributions:**

The paper provides a good dataset for OCSR. OCSR is a very useful tool for abstracting data from literature automatically. However, there are many challenges in this area, such as many abbreviations in chemistry and different written styles for different journals. This paper provides a very good dataset and also a good data augmentation method to generate synthetic data. That is the most contribution. However, there are other cases need to be considered for curating a more thorough dataset.

---

### Official Review · Reviewer_fY45 · 2022-07-21
**An awesome large-scale benchmark making great contributions to the research area of optical chemical structure recognition**

**Rating:** 8
**Confidence:** 4
**Clarity:** This paper is well written and organi…

**Strengths:**

(+) The proposed dataset is the first one that provide expert annotated labels of atoms-bonds positions in molecule images. It is a really large progress over previous datasets and provides much convenience for researchers who want to apply object detection algorithms in computer vision to the challenging optical chemical structure recognition problem.
(+) The proposed benchmark provides a large synthetic molecular image dataset for the study of transfer learning, and demonstrate the effectiveness of this strategy. It provides a promising solution for people who want to develop an optical chemical structure recognition model with small dataset, that is, first pre-training on the provided large dataset then fine-tuning on the small dataset.

**Weaknesses:**

(-) As stated in the section 5 of the paper, some categories of special chemical structures are lacking in the current dataset. Authors may try to make the dataset more comprehensive by adding those structure categories to the dataset in the future.

**Additional Feedback:**

N/A

**Correctness:**

Both the dataset construction and baseline evaluation are sound and solid. All claims made in the paper are well supported by solid experimental results.

**Documentation:**

Very detailed information about the collected dataset is provided in the supplementary file. Sufficient details are provided for reproducing the baseline methods.

**Ethics:**

No ethical review is needed.

**Relation To Prior Work:**

The differences between the proposed CEDe benchmark and previous benchmarks for optical chemical structure recognition are clearly discussed.

**Summary And Contributions:**

This work proposes CEDe, a novel and large-scale benchmark for optical chemical structure recognition, including expert labeled data, synthetic data, and multiple baseline methods. Comprehensive experiments are conducted and results show that detection-reconstruction based baselines can achieve the best performance.

---

### Official Review · Reviewer_1xhu · 2022-07-23
**Successfully reframing chemical structure identification tasks and providing a comprehensive dataset to support it**

**Rating:** 6
**Confidence:** 4

**Strengths:**

The contributions of this paper have potentially significant impact in the field of chemical entity structure recognition, which could have downstream influence on related fields such as drug discovery. The authors clearly understand the problems that prevented previous ML models from performing well and successfully argue for reframing the problem as a detection-reconstruction task. The database they introduce is large and varied, which should facilitate its use as well as further research on detection-reconstruction models for structure recognition.

**Weaknesses:**

1. I do not understand why the authors benchmark the performance of the four CNNs cited as references [31 - 34] rather than the deep learning methods developed specifically for OCSR introduced in the background and cited as references [3, 9, 10]? There is no motivation for this decision and it is not clear to me why these are appropriate benchmarks.

2. Additionally since the results of these benchmarks are used to support the claim that much fewer data can be used to perform OCSR when employed as detection-reconstruction task rather than an image-to-SMILES task, it is undecided to what extent this claim would still hold when the other models are applied to the task rather than the generic ones currently employed in the text.

3. The section on “Existing OCSR Datasets” contains background or repeated information rather than a description of the existing OCSR datasets. This section needs to be clarified: the existing datasets and their contents need to be clearly described, and their use in constructing the presented dataset needs to be articulated.

4. The sentence on line 165 remarking that ‘perfect’ graph reconstructions can be generated is strongly overstated given that the next sentence reveals that there are possibly missed or mislabeled instances.

5. The description of the dataset generation process in Section 3.1 is confusing. I could not follow it. I would suggest adding a high-level description of the sequence of steps involved in the dataset generation at the beginning of the section. I also suggest pulling out all of information describing what role the expert annotators played in producing the labeled data into one section.

6. It is not clear what impact this dataset would have on researchers working outside the field of chemical structure recognition.

**Additional Feedback:**

This contribution that has the potential for significant impact on furthering graph-reconstruction models for structure recognition, which is a field where high-quality data is difficult to come by. The writing can be confusing at times, but the contribution of the authors is clearly identifiable and appreciated. In addition to addressing the weaknesses above, the manuscript would be much improved with a comprehensive revision focused on improving grammar and readability.

Line 245: “Fine-tuned with synthetic data” should be “trained with synthetic data”

**Clarity:**

The arguments put forth by the authors in the background are easy to identify and follow. The main contributions of their work are clearly discernable and listed out.

The description of the dataset generation, however, is confusing and required multiple re-readings in order to understand.

**Correctness:**

The dataset is constructed in a sound way. The claims made by the authors concerning the data efficiency of detection-reconstruction methods compared with image-to-SMILES methods are dubious considering that this claim is supported by benchmark data from methods that appear irrelevant to ML models specifically developed for image-to-SMILES tasks.

**Documentation:**

The authors have conformed to commonly accepted conventions in the field of OCSR for the data in their dataset, and these formats are cited appropriately and described in the main text. They plan to release the dataset and the pre-trained models used for benchmarking publicly on a website as well as a Github repository and have a plan for maintenance. Currently the work cannot be reproduced since the Github repository is not active.

**Ethics:**

There are no easily conceivable ethical concerns with this dataset.

**Relation To Prior Work:**

The prior work concerning existing OCSR techniques and their shortcomings is adequately discussed. The section on existing OCSR databases is lacking as mentioned in weaknesses above.

**Summary And Contributions:**

The authors introduce a large dataset of annotated chemical entities for the purposes of advancing research into efficient machine learning techniques applied to the task of optical chemical structure recognition. The authors argue that graph representations of the chemical entities are necessary information for training models to disambiguate stereoisomeric structures or to learn about special long-range behaviors, like aromaticity. To this end the authors release the CEDe dataset, which contains around 10,000 expert-annotated chemical entities pulled from the literature and 1 million synthetically generated examples. Additionally, the authors perform benchmarking as well as introduce benchmarks suited specifically to the detection-reconstruction task they argue for in their paper.

Overall this is a good paper and I would recommend it for acceptance after a few weakness have been addressed concerning the clarity of the written text and the benchmark design choices.

---

### Official Review · Reviewer_CVtf · 2022-07-27
**A large scale chemical expression dataset with atom-level entity annotations**

**Rating:** 7
**Confidence:** 4
**Correctness:** The dataset is built in a sound way.
**Clarity:** This paper is well written and well o…

**Strengths:**

(1). The atom-bond level positional information within the image and their corresponding chemical information is provided in the CEDe dataset . This may be the first one that contains such kind of atom-level annotation information for OCSR.

(2). The proposed CEDe consists of a set of more than 700,000 bounding box annotations with carefully designed labels for chemical entity identification and molecular graph reconstruction. This provide a good benchmark for the research of OCSR field.

(3). A new synthetic dataset that  contains 1 million molecular images is released to motivate exploration of pre-training strategies for chemical entity detection and recognition.

(4).  A codebase for the synthetic data generation will be made publicly available.



**Weaknesses:**


The backgrounds of the four chemical datasets used are all relatively clean and simple, more challenging contexts and scenarios should be considered, such as camera-based chemical images with complex backgrounds, noise, and etc.

**Additional Feedback:**

What’s is the performance metric for the results reported in Table 2? Is it the exact graph matching accuracy? Please give a more detailed description on this metric. Furthermore, I am wondering how to compute matching accuracy for CNN+Transformer decoder, since it seems that this model cannot reconstruct the molecular graph directly, right? Please clarify.

**Documentation:**

Enough details about the dataset is given.

**Ethics:**

There is no ethical concern.

**Relation To Prior Work:**

Yes, it is clearly discussed how this work differs from existing related works.

**Summary And Contributions:**

This papers presents a collection of chemical expression dataset, named "Chemical Entity Detection" (CEDe), with chemical entity bounding boxes manually annotated by experts.  A chemical-entity label set (atoms, bonds, charges, pseudoatoms, etc.) is designed that covers all the necessary information for perfect compound structure identification from its constituents. These labels allow for the reconstruction and disambiguation of complex cases, such as stereoisomers, which need information that might not be explicitly present on each molecular image. In additional, a synthetic dataset containing 1 million molecular images with atoms and bonds positions within the image and their respective labels is released. Baseline benchmarks using existing popular OCSR architectures are given.

---

### Official Review · Reviewer_abjB · 2022-07-27
**Good approach on tackling the problem but the dataset and results are missing and could be polished.**

**Rating:** 5
**Confidence:** 4
**Clarity:** The writing is clear and is easy to r…

**Strengths:**

1.	The detection approach instead of the captioning approach seems to perform better and requires less amount of data
2.	Detection annotations are provided for real world datasets and a synthetic dataset is also made available (Claimed but not fulfilled).
3.	Coco annotations make it easy to use it directly with multitude of open sourced detection approached out there.


**Weaknesses:**

Many of the claims made are poorly supported
1.	The dataset and code aren’t available at the time of writing this review and cannot verify the validity.
2.	The bounding box annotations are human annotated for real world datasets which can include noise. Have the authors considered any post processing steps to nullify this? For example, tightening the bounding boxes around the text. How do the human annotations compare to the synthetic annotations?
3.	They provide no splits for the data. This can hinder any comparisons for future work using this data.
4.	Limited experiments. Please see the Additional feedback section for suggestions.
5.	There is no clear explanation of how do you go from the detection of atoms and bonds to SMILEs (although a model diagram is available). How are the pseudoatoms handled during this process. And how are the novel pseudoatoms handled that are not seen in the training data?



**Additional Feedback:**

1.	Explore other metrics for comparison. For example, the object detection metrics on the annotated data would a useful addition to understanding the failure cases.
2.	Quantify the effect of the augmentations applied on the synthetic data.
3.	Quantify the effect of pretraining. The experiments only show the performance when trained on synthetic and then finetuned on real world data. They mention that they have tried using only real world data but don’t report the results. These results can actually highlight the importance of the synthetic data.
4.	In table 2, the column “Dataset type” is a bit misleading. Maybe “Training dataset” is a better column name?


**Correctness:**

The claims arent't fully corroborated and have issues as discussed in other sections.

**Documentation:**

The dataset link points to only a sample of the CEDe dataset and there are no examples of the CEDe synthetic dataset. They also promise to release the code for the data generation which is not available. The code for the experiments is also not available which makes the experiments irreproducible.

**Ethics:**

No concerns

**Relation To Prior Work:**

Prior work is discussed well and the improvements are also well established.

**Summary And Contributions:**

The paper introduces CEDe, a dataset of chemical structure annotations. For real world datasets, they add annotations to the already available open source OCSR datasets (UOB, USPTO, CLEF, JPO). They also introduce a synthetic dataset from a million compounds from PubChem to facilitate pretraining. To this end, they develop specific augmentations that could improve the transfer from synthetic to real world data. The annotations in these datasets consist of atom, bond level labels to facilitate detection approaches. They frame the problem of detecting chemical structures as detecting the atoms and bonds and then recreating the graph from these predictions. They show that this approach beats the baselines most of the time when using exact match as a metric. They also show that this approach requires significantly less data than the image to Smiles method.

---

### Official Review · Reviewer_kC2V · 2022-07-27
**No actual dataset, no proper review**

**Rating:** 5
**Confidence:** 3

**Strengths:**

* Possibly high-quality annotation
From the example shown in the manuscript, the annotation quality is quite high.

**Weaknesses:**

The dataset is not yet to reviewers; the link to Github, which is claimed to contain the baseline implementation, is 404 in the review period. It is not even included in supplementary materials. Therefore, I concluded that the dataset availability and accessibility are not met.

**Additional Feedback:**

The dataset should be disclosed at least to the reviewers.

**Clarity:**

The reviewer found no problem to understand the manuscript.


**Correctness:**

The reviewer thinks the dataset is constructed properly.


**Documentation:**

Cannot judge since the dataset is not disclosed even to the reviewers.

**Relation To Prior Work:**

The authors clearly disccussed the demand for the annotation of the dataset, and claimed to have annotated them.


**Summary And Contributions:**

The authors claim to have annotated the Optical Chemical Structure Recognition dataset. Regarding the example in the manuscript, the annotation format is well thought out, and making the chemical structure machine-readable is an essential issue for contributing to the further development of MI, and high-quality data sets are crucial for that purpose.
However, the dataset itself and the baseline implementation is not disclosed to the reviewers and cannot be evaluated.

---

### Meta-Review · Program_Chairs · 2022-09-17

**Recommendation:** Accept
**Confidence:** 3

**Metareview:**

Given the extensive discussions, it's good to make a small overview:
Reviewer kC2V ('weak reject') didn't do a full review since the data and code were not available. Both become available, afterwards, but very late in the process.
Reviewer abjB ('weak reject' as a lower bound) raised many issues about limitations in the experiments. Most of these were resolved by additional experiments.
Reviewer CVtf ('accept') raised minor issues about the simplicity of the chemical datasets. The authors added additional images as a result.
Reviewer 1xhu ('weak accept') raised many issues, especially on the correctness of claims. The authors addressed most of them to some extend.
Reviewer fY45 ('clear accept') raised minor issues about adding more chemical structures
Reviewer gkZt ('weak reject') also raised minor issues about adding more chemical structures. The authors resolve some, others seems to be misunderstandings.
Overall, it seems that the balance learn toward acceptance. The authors did a lot of work to achieve this. Much confusion would have been saved if the data and code was shared correctly from the start.

---

### Decision · Program_Chairs · 2022-09-17

Accept